# Cryo-EM structures of mitochondrial ABC transporter ABCB10 in apo and biliverdin-bound form

Sheng Cao[1,3], Yihu Yang[1,3], Lili He[2], Yumo Hang[2], Xiaodong Yan[1], Hui Shi[1], Jiaquan Wu[1] & Zhuqing Ouyang [2] ✉

ABCB10, a member of ABC transporter superfamily that locates in the inner membrane of mitochondria, plays crucial roles in hemoglobin synthesis, antioxidative stress and stabilization of the iron transporter mitoferrin-1. Recently, it was found that ABCB10 is a mitochondrial biliverdin exporter. However, the molecular mechanism of biliverdin export by ABCB10 remains elusive. Here we report the cryo-EM structures of ABCB10 in apo (ABCB10-apo) and biliverdin-bound form (ABCB10-BV) at 3.67 Å and 2.85 Å resolution, respectively. ABCB10-apo adopts a wide-open conformation and may thus represent the apo form structure. ABCB10-BV forms a closed conformation and biliverdin situates in a hydrophobic pocket in one protomer and bridges the interaction through hydrogen bonds with the opposing one. We also identify cholesterols sandwiched by BVs and discuss the export dynamics based on these structural and biochemical observations.

Mitochondria are double-membrane-bound organelles and major cellular supply of adenosine triphosphate (ATP) with harmful byproducts reactive oxygen species (ROS). Superoxide and hydrogen peroxide ($H_2O_2$), which are the main form of ROS in mitochondria can damage cells and have to be cleared away. A known $H_2O_2$ scavenger is bilirubin (BR), which can be oxidized to biliverdin (BV). BV is then exported to cytosol where it is reduced back to BR by biliverdin reductase (BLVRA)[1]. BR can cross the mitochondrial membrane by passive diffusion due to its high lipophilicity. BR is then oxidized by $H_2O_2$ and complete the cycle (Fig. 1). One key step in this cycle is the export of hydrophilic BV. Recently, it has been reported that BV is exported from mitochondria by an ATP-binding cassette transporter (ABC transporter), ABCB10[2].

ABC transporters are a superfamily of integral membrane proteins that are ubiquitously present in all organisms, from bacteria to humans. They transport substrates across membranes by utilizing the energy of ATP binding and hydrolysis[3]. ABC transporters can uptake nutrients, vitamins, trace metals and biosynthetic precursors (importers) and export substances such as sterols, metabolites, lipids and drugs (exporters)[4]. Mutations in or dysfunctions of these proteins have been implicated in diverse disorders, such as diabetes, genetic diseases, cystic fibrosis, hypercholesterolemia, retinal degenerations, lipid trafficking disorders cystic fibrosis and adrenoleukodystrophy[5]. ABC transporters are also involved in multidrug resistance of tumor cells as they can extrude chemotherapeutic agents[6,7]. Most ABC transporters, especially exporters function as a homodimer. Each monomer contains two parts, the transmembrane domains (TMDs) and the nucleotide binding domains (NBDs). The NBDs that are located in the cytoplasm and responsible for ATP binding and hydrolysis are very conserved. While the TMDs that form the translocation pathway are variable in order to accommodate the diversity of substrates[8–10]. An alternating access mechanism has been proposed for ABC transporters where the substrate-binding sites switch between outward- and inward-facing conformations[4,11].

ABCB10 is a member of ABC transporter superfamily that locates in the inner membrane of human mitochondria, with the NBDs within the mitochondrial matrix[12]. It was first identified by screening genes regulated during erythroid differentiation by the transcription factor

[1]Wuxi Biortus Biosciences Co. Ltd., 6 Dongsheng Western Road, 214437 Jiangyin, Jiangsu, China. [2]Department of Pathogen Biology, School of Basic Medicine, Tongji Medical College, Huazhong University of Science and Technology, 13 Hangkong Road, 430030 Wuhan, Hubei Province, China. [3]These authors contributed equally: Sheng Cao, Yihu Yang. ✉e-mail: zhuqingouyang@hust.edu.cn

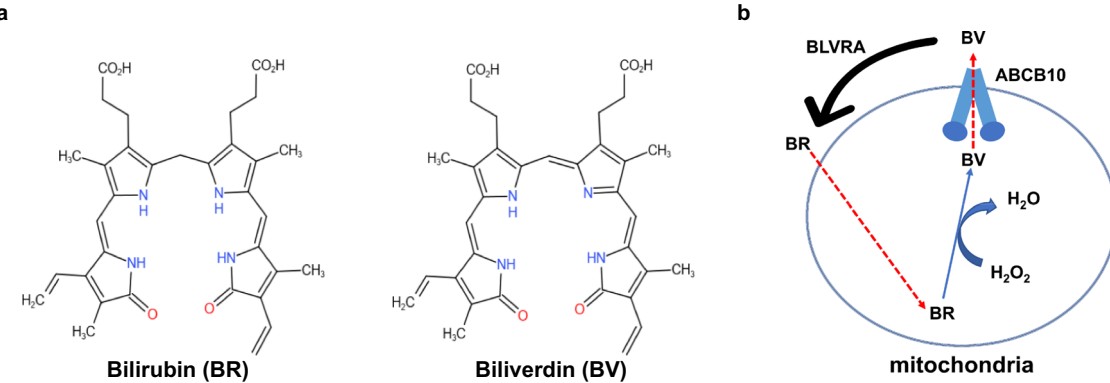

**Fig. 1 | The biliverdin (BV)-bilirubin (BR) cycle. a** Chemical structures of BV and BR. **b** Model representation of the BV-BR cycle. BV is exported by ABCB10 to cytosol where it is reduced into BR. BR can cross the mitochondrial membrane passively due to its high lipophilicity then oxidized by $H_2O_2$.

GATA-1 and named ABC-me (ABC-mitochondrial erythroid)[13]. It was found later to play crucial roles in antioxidative stress and stabilization of the iron transporter mitoferrin-1[14–16]. ABCB10 is essential for hemoglobin synthesis and erythropoiesis and ABCB10 knocked out mice die at day 12.5 of gestation[17,18]. ABCB10 deletion in liver protects obese mice from insulin resistance, steatosis and hyperglycemia. Multidrug resistance-like 1 (Mdl1), the yeast homolog of ABCB10, was identified to transport peptides with molecular masses of ~600 to 2100 daltons[19]. It is therefore hypothesized that ABCB10 is a peptide transporter[20]. Recently, Shum et al. demonstrated that the substrate of ABCB10 is biliverdin (BV). However, whether biliverdin can interact with ABCB10 directly and how ABCB10 exports biliverdin out of the mitochondria is unclear.

In this work, we report the cryo-EM structures of ABCB10 in apo and biliverdin-bound form. ABCB10-BV forms a closed conformation and biliverdin situates in a hydrophobic pocket in one protomer and bridges the interaction through hydrogen bonds with the opposing one. We also identify cholesterols sandwiched by BVs and predict cholesterol may serve as a cofactor of BV export by ABCB10.

## Results
### Characterization and structure determination of ABCB10-apo and ABCB10-BV

To solve the structure of human ABCB10, we expressed the protein in mammalian Expi293F Gn-TI- cells without the N-terminal mitochondrial targeting sequence. The construct contains ABCB10 (residue 152–738) with C-terminal flag and 10x histidine affinity tags. We also made the glutamate to glutamine (E659Q) mutation in Walker-B motif, which could strongly reduce the ATPase activity and allow trapping the ABC transporters in a pre-hydrolytic state. Both ABCB10 wild type (WT) and E659Q were purified in glyco-diosgenin (GDN) plus cholesteryl-hemisuccinate (CHS). The proteins appear to be stable and homogeneous after extraction and purification with anti-flag affinity chromatography and size-exclusion chromatography (Supplementary Fig. 1a). To test whether the proteins are functional, the ATPase activity of purified ABCB10 were examined. As expected, ABCB10-WT hydrolyzes ATP efficiently while E659Q devoid of ATPase activity, suggesting that the purified ABCB10 is a functional ATPase (Supplementary Fig. 1b). We also performed the thermo shift assay to test whether the purified ABCB10 proteins could be stabilized in the presence of biliverdin IX alpha (BV). It was confirmed that BV could bind to the detergent extracted ABCB10 in a concentration dependent manner. The melt temperature (Tm) of ABCB10 is elevated after incubation with BV (from 44.85 °C to 47.35 °C) and the apparent affinity was ~8.78 μM. These results suggest that that BV can bind and stabilize ABCB10 (Supplementary Fig. 1c, d).

To determine the BV-bound ABCB10 structure, hopefully in outward-facing conformation, to uncover the molecular mechanism of BV export, ABCB10 E659Q mutant protein was vitrified in the presence of 1 mM BV. Unexpectedly, we obtained both BV-free (ABCB10-apo) and BV-bound (ABCB10-BV) structures at 3.67 Å and 2.85 Å resolution cryo-electron microscopy (EM) maps from a single data set after data processing, respectively (Supplementary Fig. 2). We used the structure of ABCB10 bound to AMPPCP (PDB ID: 4AYX) to guide the model building. The final model of ABCB10-apo encompasses residues from 156 to 724, missing the C-terminal 14 residues and 528-532 in NBD. ABCB10-BV contains residues from 154 to 724, missing the C-terminal 14 residues and 629-631. The majority of the side chains in the TMDs of both structures can be clearly defined (Supplementary Figs. 3 and 4; Table 1). The EM density maps of ABCB10-BV are sufficiently clear to place the bound BV (Supplementary Fig. 4). The structure solution methods are summarized and described in details in supplementary materials and Methods.

**Table 1 | Cryo-EM data collection, refinement, and validation statistics**

| | ABCB10-apo | ABCB10-BV |
|---|---|---|
| **Data collection and processing** | | |
| Magnification | 105,000 | |
| Voltage (kV) | 300 | |
| Dose rate (e-/(pix·s)) | 14.81 | |
| Pixel size (Å/pix) | 0.85 | |
| Total dose (e-/Å²) | 55.21 | |
| Defocus range (μm) | −1.1 to −1.8 | |
| Particle images | 70,134 | 206,922 |
| Map resolution (Å) | 3.67 | 2.85 |
| **Model refinement and validation** | | |
| Initial PDB model | 4AYX | 4AYX |
| Non-hydrogen atoms | 4169 | 4324 |
| Protein residues | 565 | 567 |
| Mean B factor-protein | 43.5 | 58.0 |
| Mean B factor-ligand | - | 30 |
| R.m.s deviations of bond lengths (Å) | 0.003 | 0.005 |
| R.m.s deviations of bond angles (°) | 0.983 | 1.229 |
| MolProbity score | 1.52 | 1.50 |
| Clash score | 5.03 | 7.99 |
| Ramachandran outliers | 0.00 | 0.00 |
| Ramachandran allowed | 3.76 | 2.31 |
| Ramachandran favored | 96.24 | 97.69 |

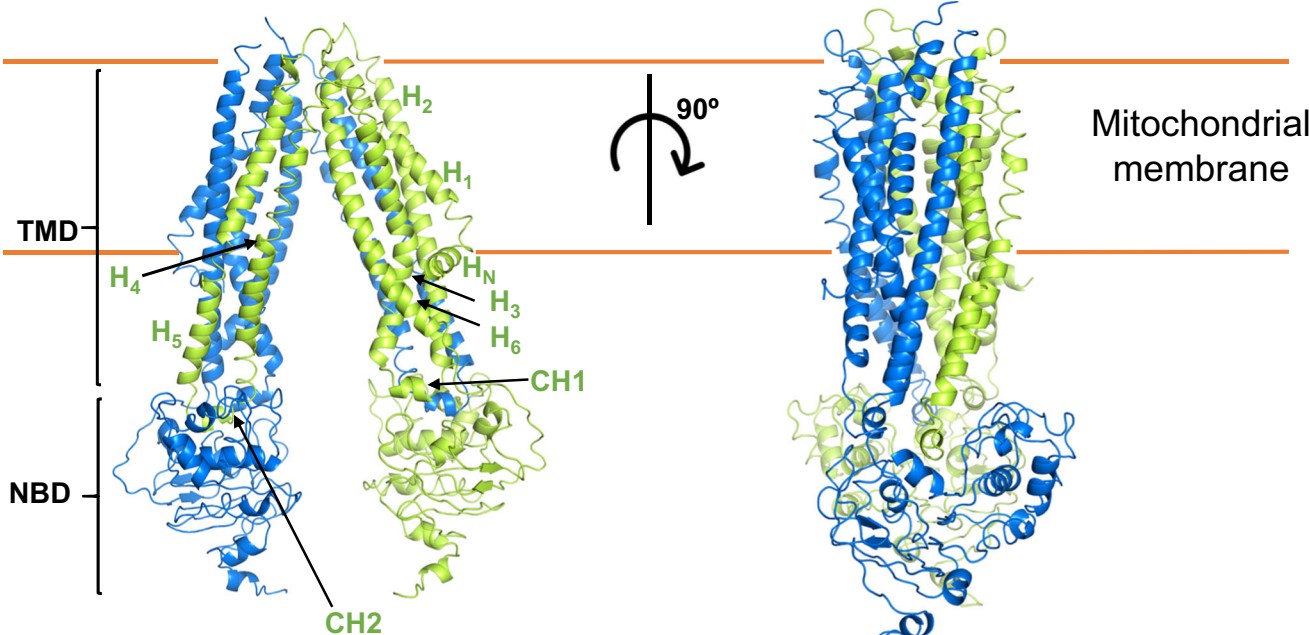

**Fig. 2 | Structure of BV exporter, ABCB10-apo.** Structure of ABCB10-apo in two different orientations. Cartoon drawing of ABCB10-apo. ABCB10 forms an inverted V-shape homodimer, each monomer (colored in marine and limon, respectively) contains Transmembrane Domain (TMD) and nucleotide binding domain (NBD). The N-terminal helix ($H_N$), TMH 1-6 (H1-H6) and coupling helices (CH1 and CH2) in one protomer are labeled.

## Structures of ABCB10-apo and ABCB10-BV

ABCB10-apo forms an inverted V-shape homodimer structure from the side view via the TMDs and each protomer is similar to that of ABCB10 in nucleotide-free form (ABCB10-NF, PDB ID: 3ZDQ) as previously reported[21]. Each protomer contains a short N-terminal helix ($H_N$), followed by six long transmembrane helices (TMH1-TMH6) and a conserved NBD connected by a long linker. The N-terminal helix is near parallel to the membrane surface. There are two small helices located at TMH2-TMH3 and TMH4-TMH5, respectively, which are called coupling helices (CH1 and CH2) as they mediate the interaction between TMDs and NBDs. The two protomers are tethered together by extensive interactions between TMD4-TMD5 from one protomer and TMD1-3 and TMD6 from the opposing one. ABCB10-apo dimer adopts an inward-facing, wide-open conformation, awaiting the transport substrate (Fig. 2). ABCB10-BV exhibits inverted V-shape homodimer and inward-facing conformation as well. The secondary structures of ABCB10-BV are well defined compared to that of the ABCB10-apo where most β-strands in NBDs are recognized as loops, probably due to the low local resolution (Supplementary Figs. 3 and 4). A close view of the translocation pathway of substrates reveals a mostly hydrophobic pocket with some small patches of positive charge composed of R295 and R232, indicating that it prefers substrates with hydrophobicity (Supplementary Fig. 5). Interestingly, two cholesterol molecules are sandwiched between two BVs.

## Structural comparison of ABCB10

Previously Shintre et al. determined the crystal structures of ABCB10 in nucleotide-free form (ABCB10-NF, PDB ID: 3ZDQ) and in ATP analog bound form (ABCB10-AMPPNP, PDB ID: 4AYW)[21]. They found that these two structures are very similar and both adopt an inward-facing conformation. These findings are contradictory to other ABC transporters, which exhibit an outward-facing conformation when bound to ATP. To reveal the conformational changes after substrate and ATP binding, we compared the four structures. All four structures adopt an inverted V-shape dimer conformation. However, ABCB10-apo exhibits a more wide-open conformation than other three (Fig. 3a), suggesting that either ATP or BV can trigger the conformational changes. ABCB10-

apo holds a much bigger pocket for substrate, indicating that it is more competent for substrate-binding than others. ABCB10-apo may thus represent the initial stage of substrate export. Interestingly, ABCB10-NF that contains no substrate or ATP analog adopts a closed conformation as well. This structure may reflect one of the intermediate conformations, while other factors may be able to trigger the conformational changes. Intriguingly, we noticed that ABCB10-NF contains cardiolipin and dodecyl-beta-D-maltoside in their TMDs. Crystal packing might be another factor responsible for these changes.

Export of substrate requires ATP binding and hydrolysis and ATP is sandwiched between the Walker A and Walker-B motifs from one NBD and the signature motif from the other. To gain insights into the detailed conformational changes we inspect the distance and relative orientation between two NBDs. The ATPase sites are about 30 Å apart in apo form, a distance far beyond the reach of ATP molecules, which is consistent with the low basal ATPase activity observed before[21]. Binding with BV bring the two NBD close to about 13 Å (Fig. 3b). We therefore believe that the ATPase activity of ABCB10 can increase dramatically in the presence of BV, consistent with an observation reported before[2].

ATP and substrate binding induce rearrangement of TMHs and eventually bring about an outward-facing conformation. ABCB10-NF, -AMPPNP and -BV represent several intermediate conformations during this transition. To reveal the rearrangement of TMHs in details we did superimposition of the monomeric ABCB10 structures. The $H_N$, TMH1-3 and TMH6 in these structures align well with each other while TMH4 and TMH5 have a big conformational change. Compared to those in ABCB10-apo, these two helices in other three structures shift toward TMH1-3 and TMH6 to form a more closed conformation. Among the three, ABCB10-BV shows the most prominent shift, while shifts in ABCB10-NF and ABCB10-AMPPNP are comparable, with the former slightly more prominent (Fig. 3c).

## Structure interface of ABCB10-BV

The resolution of cryo-EM map of the BV-bound form is much higher than that of the apo form (2.85 Å versus 3.67 Å), suggesting that BV binding stabilizes the transporter, which is consistent with the thermo shift assay results. The densities in the TMD are particularly well

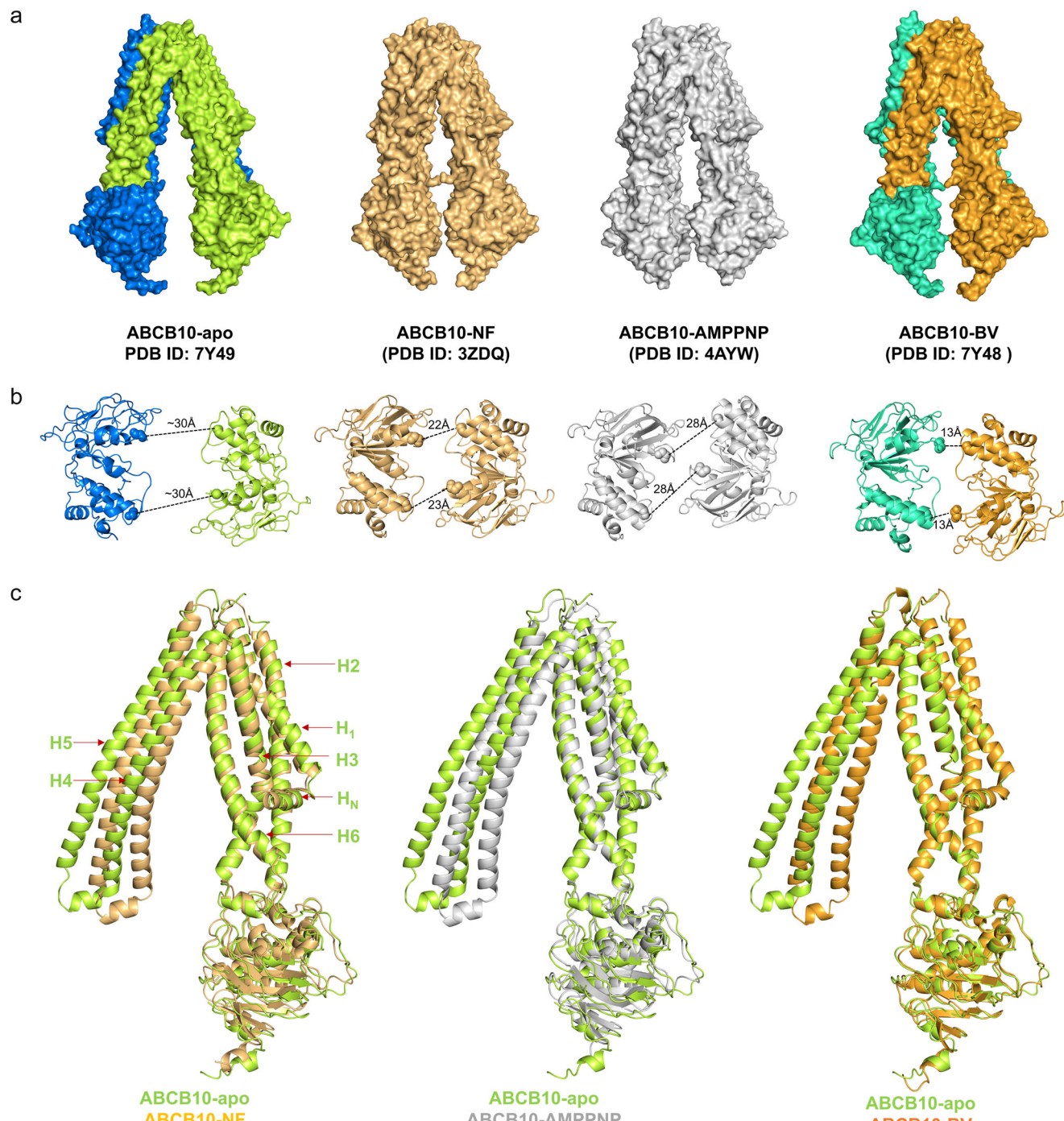

**Fig. 3 | Structure comparison of ABCB10. a** Surface drawing of ABCB10-apo, nucleotide-free form (ABCB10-NF), ATP analogs bound form (ABCB10-AMPPNP) and biliverdin-bound form (ABCB10-BV). **b** Alignment of NBDs of the structures shown in (A) viewed looking toward the membrane. The two NBDs, viewed from the cytoplasm along the membrane normal. The Cα distances between the Walker A glycine (Gly527) and the serine (Ser635) in the signature motif are indicated. **c** Monomeric ABCB10-NF, -AMPPNP, and -BV structures are superimposed with ABCB10-apo separately.

resolved, enabling us to define the position and the identity of substrate unambiguously (Supplementary Fig. 4).

The ABCB10 full transporter binds two BV molecules simultaneously. Two symmetric BV-binding sites locates in the middle of the inner membrane, more than half way to destination (Fig. 4a). BVs adopt a nearly planar configuration that is vertical to the mitochondrial inner membrane (Fig. 4b). Each BV molecule is sandwiched at the interface of two protomers. TMH1-2 from one protomer and TMH4-6 from the opposing one form a pocket for one substrate (Fig. 4b). A close view reveals that BV lies in a hydrophobic pocket (Supplementary Fig. 5)

with two carboxyl groups pointing outwards. Indeed, the interactions between BV and ABCB10 can be divided into two parts: The hydrophobic interaction with one protomer and the hydrogen bond interaction with the other. BV contacts with residues on TMH4 (V322, V325, S326 and A329), TMH5 (F398, F399, T402, S405, N407, I409, and V410) and TMH6 (F439 and I443) hydrophobically. Two residues from the opposing protomer, namely S185 on TMH1 and N229 on TMH2, form hydrogen bonds with two carboxyl groups of BV (Fig. 4c). The small patches of positive charge composed of R295 and R232 also contribute to the BV interaction, although either of them binds weakly to the

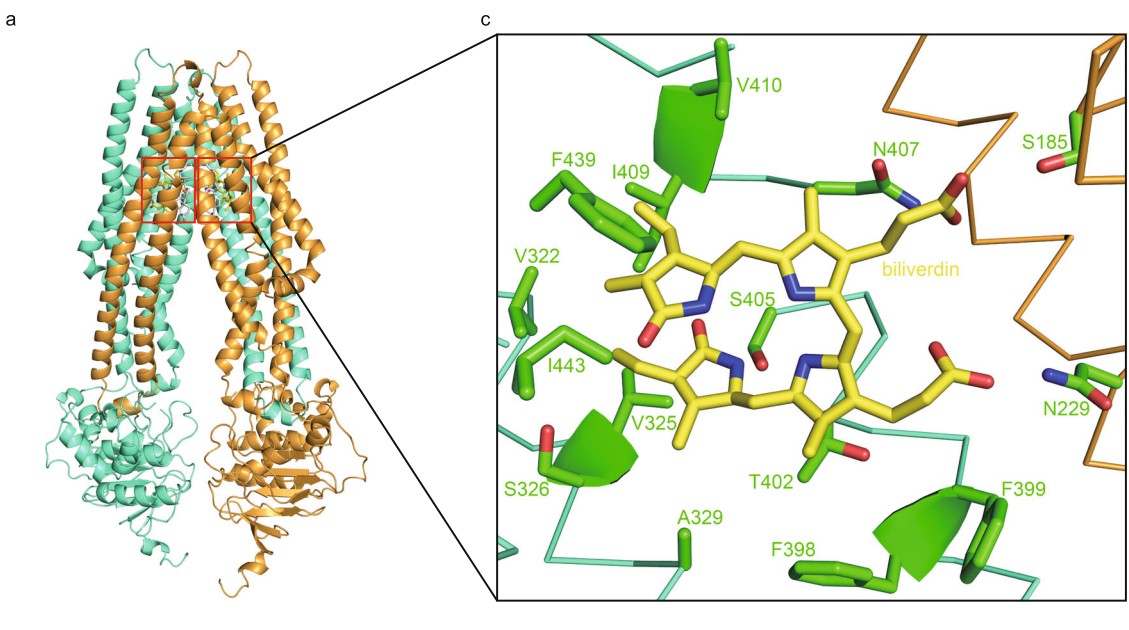

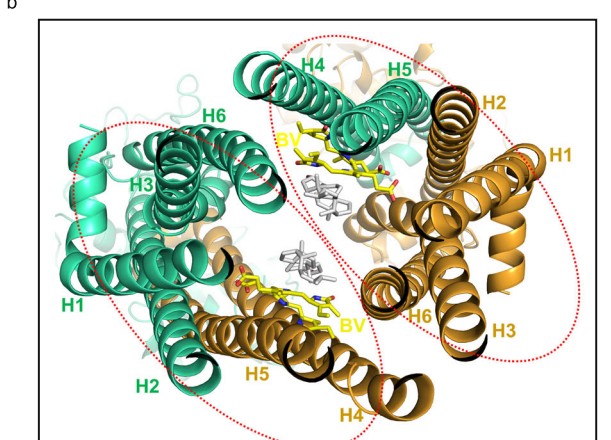

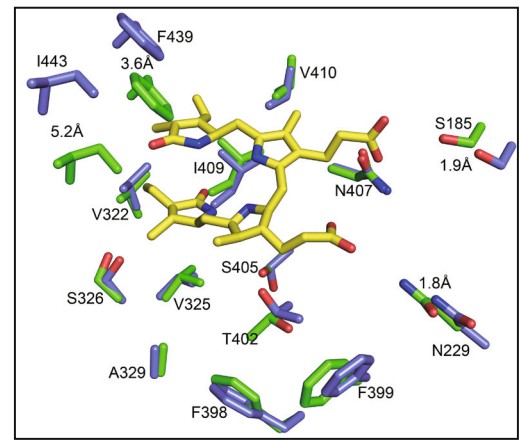

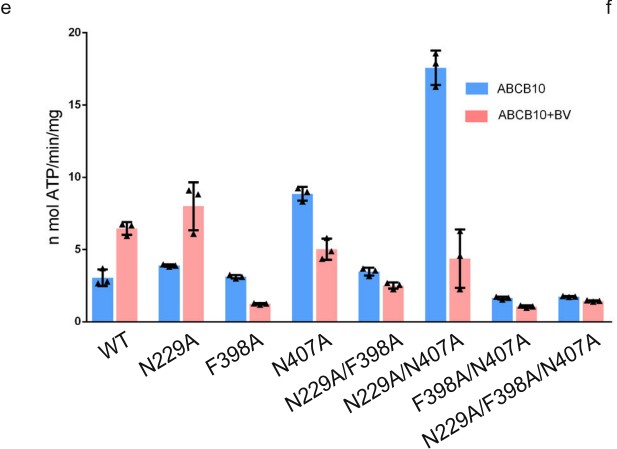

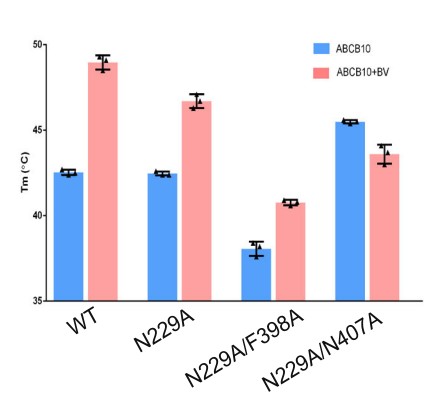

carboxyl group of BV (The distances between these residues and the carboxyl group of BV are 4.1 Å and 5.8 Å, respectively). Structural comparison of ABCB10 in the apo and BV-bound form reveals local conformational changes induced by substrate binding. Residues on TMH4 and TMH5, including V322, V325, S326, A329, F398, F399, T402, N407, I409, and V410 have minor conformational changes. Interestingly, S405 flips its side chain upon substrate binding. However, residues on TMH6 and TMH1-2 of the other protomer show remarkable

shifts. F439 and I443 on TMH6 shift 5.2 Å and 3.6 Å, respectively, toward BV and residues on TMH1 and TMH2 (S185 and N229) move about 2 Å toward the substrate (Fig. 4d). Since residues on TMH4-5 form the major binding sites and this surface is far away from other BV-contacting residues in ABCB10-apo structure, we speculate that BV may prime to ABCB10 via TMH4-5 first and then bridge the interaction with TMH6 and TMH1-2 from the other protomer, thus trigger the conformational changes to a more closed one. Interestingly, several of

**Fig. 4 | Interactions of BV with ABCB10. a** The ABCB10 full transporter binds two biliverdin molecules simultaneously. Cartoon drawing of ABCB10-BV with side view showing the substrate-binding sites. Biliverdins are colored in yellow and boxed. **b** Top view of the interface between ABCB10 and BV. Biliverdins and cholesterols are shown in sticks and colored in yellow and white, respectively. BV is sandwiched at the interface of two protomers. TMH1-2 (H1-H2) from one protomer and TMH4-6 (H4-H6) from the other one form a pocket for one biliverdin. Cholesterols are sandwiched by two BV molecules. **c** Zoom-in view of the interface between ABCB10 and BV. Residues that involved in BV interaction are labeled. Biliverdins are shown in sticks and colored in yellow. Hydrogen bonds between BV and N229 or S185 are indicated. **d** Local conformational changes at the substrate-binding site. Residues on TMH4 (V322, V325, S326, and A329) and TMH5 (F398, F399, T402, S405, N407, I409, and V410) have minor conformational changes while residues on TMH6 (F439 and I443), TMH1 (S185) and TMH2 (N229) show major shifts. **e** ATPase activity of ABCB10 WT and mutants. ATPase activities are measured at varied concentration of ABCB10 WT or mutant. Each data is presented as the means±s.d. of three independent assays ($n = 3$). **f** BV influences the thermal stabilities of ABCB10 WT and mutants. The melting temperatures (Tm) of ABCB10 are measured by performing thermofluor shift assays. Each data is presented as the means ± s.d. of three independent assays ($n = 3$).

these BV-contacting residues, including N229, A329, R232, R295, F398, and N407 are strictly conserved and S185, V322, V325, F399, I409, and V410 are loosely conserved as well from yeast to human (Supplementary Fig. 6), suggesting that BV binding is a conserved feature of ABCB10.

To validate this BV-binding surface of ABCB10, we mutated the strictly conserved residues to alanine. As expected, BV significantly stimulates the ATPase activity of ABCB10 WT. However, BV fails to enhance the ATPase activities of ABCB10 mutants except N229A (Fig. 4e). Interestingly, the ATPase activities of ABCB10 N407A and N229A/N407A increase significantly compared to that of ABCB10 WT. These mutants may undergo a conformational change to a more closed one that favors ATP binding. Consistently, BV increases the thermal stability of ABCB10 WT much significantly than that of ABCB10 mutants (Fig. 4f).

### Cholesterol is a cofactor of ABCB10-BV

During the model building process, we unexpectedly noticed that two cholesterol molecules are sandwiched between two BVs (Supplementary Fig. 4c). Cholesterols run parallel to BVs with the hydroxy group pointing to the lumen of mitochondria and form extensive hydrophobic interactions with adjacent BV molecule (Figs. 4b and 5a). Cholesterols also contact with ABCB10 directly via hydrophobic interaction with F439, V410 and I443, residues that are also involved in BV interaction. There are also weak hydrophobic interactions between two cholesterol molecules. Since cholesterol has similar structure with CHS, the detergent to solubilize ABCB10, we measured the ATPase activity and Tm of ABCB10 in the presence or absence of cholesterol to confirm its identity. Cholesterol promotes the ATPase activity of ABCB10 in a dose-dependent manner. The ability of cholesterol to stimulate the ATPase activity of ABCB10 mutants such as N229A, F398A, N407A and N229A/F398A is compromised or abolished (Fig. 5b, c). Both cholesterol and BV enhance the thermal stability of ABCB10 WT or N229A. Incubation with two compounds simultaneously displays synergic effect (Fig. 5d). These results confirm the identity of the ligand in ABCB10-BV structure.

### Discussion

Previously, Shintre et al. has determined the crystal structure of ABCB10 in nucleotide-free (ABCB10-NF) and nucleotide-bound (ABCB10-AMPPNP) states. These two structures show minor differences and both in inward-facing, closed conformation[21]. Our ABCB10-apo structure which is in wide-open conformation may represent the initial stage of export, namely apo form, because it contains a much bigger pocket and is thus more susceptible to substrates. How the open ABCB10-apo transport substrates from mitochondria to cytosol? The most popular model of export is the alternating access mechanism based on the structures of MsbA[22,23] and Sav1866 from bacteria[24]. In this model, the ABC transporters switch from an apo inward-facing to a nucleotide-bound outward-facing conformation with subunit intertwining, in which TMH4-5 are swapped in the inward-facing dimer whereas in the outward-facing dimer TMH1-2 are swapped between two half-transporters. Our ABCB10-apo structure is consistent with this model in that TMH4-5 are swapped in the inward-facing dimer. However, both the previously reported nucleotide-bound form (ABCB10-

AMPPNP) and our substrate-bound form (ABCB10-BV) of ABCB10 adopt an inward-facing, closed conformation. These observations suggest that either ATP or BV can trigger the conformational changes from wide open to a closed state, but it may need both to achieve the outward-facing conformation[25], a mechanism that has been applied for ABC transporters such as McjD[26]. It makes sense considering that mitochondria are ATP producing factory with high concentration of ATP, conformational change to outward-facing state with ATP alone without substrate exporting is a big waste of energy.

How ATP binds to and drives the NBDs is well appreciated. Two nucleotides are sandwiched between two NBDs and bring together the NBDs, which triggers conformational changes of the TMDs to a more closed conformation. ATP hydrolysis separate them apart. BV binding can also trigger a similar conformation change. Since the hydrophobic patches on TMH4-5 are the major binding site for BV and the BV-contacting residues on TMH1,2 and 6 are far away in ABCB10-apo structure, we speculate that BV may prime to TMDs via TMH4-5 first, then bridge the interactions with residues on TMH6 and TMH1-2 from the opposing protomer. ABCB10 may bind ATP and BV simultaneously rather than sequentially to achieve efficient export for two reasons: (1) The NBDs in ABCB10-AMPPNP are widely separated, similar to that in ABCB10-NF, therefore binding to ATP alone is not sufficient to trigger efficient ATP hydrolysis and full conformational changes, although basal ATPase activity is observed[21]; and (2) Either ATP or BV-bound form is in closed conformation, which may impede the access of the second compound.

It takes several steps for ABCB10 to export the substrates, from the wide open, inward-facing conformation with TMH4-5 swapped to the outward-facing and probably TMH1-2 swapped conformation. BV may first prime to TMH4-5 through hydrophobic interaction and then bridge the interaction with TMH6 and TMH1-2 from the opposing one. This driving force in concert with ATP binding triggers rearrangement of TMDs into the outward-facing conformation by disrupting the BV-ABCB10 interactions and therefore release BV in the cytosol. Substrate and ATP binding are exterior factors whereas intrinsic tendencies to form intramolecular interaction and abilities to perform conformational change are interior factors for this transformation. Close inspection of sequence alignment reveals that the most conserved regions in NBDs are those involved in ATP binding and hydrolysis (Walker A motif, Walker-B motif, C motif, and H motif) and TMD binding (Q-loop). BV-contacting residues are also conserved in TMDs. We notice that some other residues are strictly conserved in TMDs, which may participate in conformational changes. Additionally, these conserved residues are glycine rich with 3 glycine residues on TMH3, 4 on TMH5 and TMH6, which may be responsible for the flexibility of TMHs[27] (Fig. S5). ATP hydrolysis drives NBDs apart, allowing the structure to return to the initial stage and complete the cycle. A snapshot of ABCB10 in outward-facing conformation is needed to fully understand the export cycle.

In the ABCB10-BV structure we observed two cholesterol molecules sandwiched by two BVs. Our in vitro assay results confirm the cholesterol identity. Since CHS is a derivative of cholesterol, it is possible that cholesterol is a product of hydrolysis of CHS. It may also come from contaminant of biliverdin or other reagents. Cholesterol

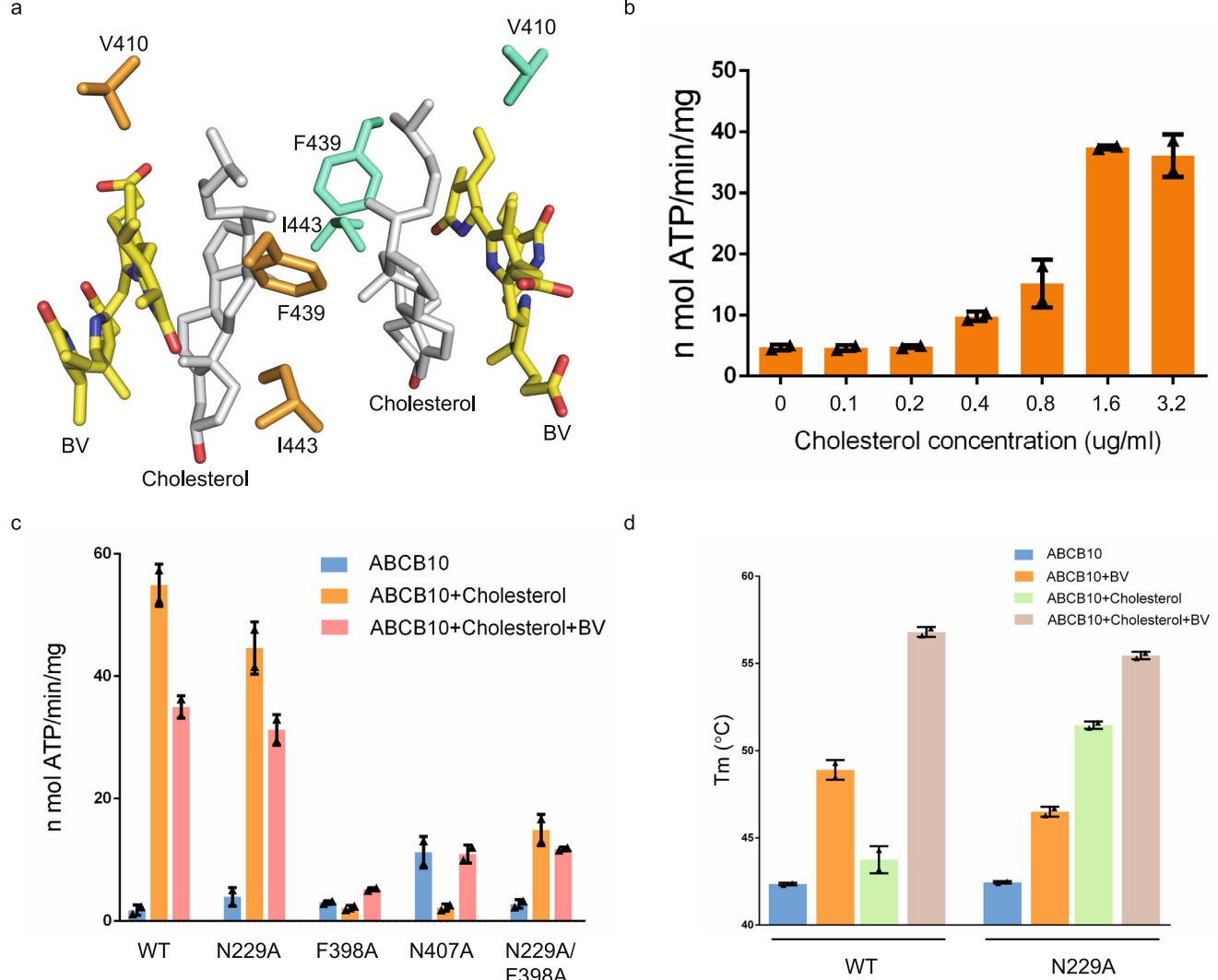

**Fig. 5 | Cholesterol is a cofactor of ABCB10-BV. a** Two cholesterol molecules are sandwiched by two biliverdin molecules. Zoom-in view of the interface between cholesterol (in gray) and BV. Residues that involved in cholesterol interaction are labeled. Biliverdins are shown in sticks and colored in yellow. **b** ATPase activity of ABCB10 with increasing concentration of cholesterol. ATPase activities are measured at varied concentration of ABCB10 WT. Each data is presented as the means ±s.d. of three independent assays ($n = 3$). **c** ATPase activity of ABCB10 WT and mutants. ATPase activities are measured at varied concentration of ABCB10 WT or mutant in the presence of absence of cholesterol and biliverdin. Each data is presented as the means±s.d. of three independent assays ($n = 3$). **d** Cholesterol influences the thermal stabilities of ABCB10 WT and mutants. The melting temperatures (Tm) of ABCB10 are measured by performing thermofluor shift assays. Each data is presented as the means±s.d. of three independent assays ($n = 3$).

interacts extensively with BV although weak interaction with ABCB10 is also observed. Export of substrate by ABC transporters consists of several sequential conformational changes triggered by ATP binding and hydrolysis, which leads to rearrangement of TMDs and disruption of the interaction between ABC transporter and substrate. Since BVs form extensive hydrophobic interaction with cholesterols and cholesterol-binding residues, e.g., F439, V410, and I443 are also involved in BV binding, the outward-facing conformation of ABCB10 may loss contact with cholesterol as well. Consequently, cholesterols might be exported from mitochondria along with BVs. These observations suggest that cholesterol might serve as a cofactor of BV export and be exported by ABCB10 simultaneously. Although it is possible that cholesterol is located there because that region of ABCB10-BV provides an environment that accommodates the interaction with hydrophobic and amphipatic molecules. It might be also the reason that cholesterol can be seen in ABCB10-BV but not in ABCB10-Apo. Both cholesterol and BV enhance the thermal stability of ABCB10 WT or N229A, while BV limits the ability of cholesterol to increase ATPase activity (Fig. 5c). It might be possible that cholesterol regulates

ABCB10 ATPase activity independently of BV transport as Shum et al. already showed BV transport without cholesterol[2]. Further studies are needed to test this highly speculative hypothesis. Cholesterol is a crucial component of membrane bilayers and mitochondrial cholesterol plays important roles in biogenesis, maintenance of their membrane, steroid hormone and bile acids synthesis. Elevated cholesterol levels are associated with cancer development[28]. Interestingly, it is reported that increased mitochondrial cholesterol levels are associated with decreased antioxidant levels[29]. It is possible that cholesterol facilitates ROS clearance via co-export with BV by ABCB10. More studies are needed to test this hypothesis.

## Methods
### Protein expression and purification
The synthetic human ABCB10 gene (residues A152-A738) was codon-optimized for mammalian cell expression. Both the wild-type protein and the E659Q mutant construct were cloned into a BacMam expression vector containing a C-terminal TEV protease site, FLAG tag and 10xHistidine tag. Baculovirus carrying either recombinant ABCB10

construct was amplified in Sf9 cells. For protein expression, Expi293F GnTI⁻ cells were infected with recombinant baculoviruses, grown at 37 °C for 8 h, induced with 10 mM sodium butyrate, and then were grown at 30 °C for 48 h before harvesting.

Cell pellets were resuspended in hypotonic buffer containing 50 mM HEPES, pH 7.5, SMNE and protease inhibitors (1 mM PMSF, 1 mM benzamidine, 0.1 μg/mL trypsin inhibitor, 0.1 μg/mL pepstatin A, 0.1 μg/mL leupeptin, 0.1 μg/mL aprotinin, and 3 μg/mL DNase). Then centrifuge at $25,000 \times g$ for 45 min after dounce 20 times. The membranes were resuspended with lysis buffer containing 20 mM HEPES, pH 7.5, 150 mM NaCl and protease inhibitors, then solubilized with 0.5% LMNG, 0.05% CHS at 4 °C for 2 h. Solubilized membranes were centrifuged at $25,000 \times g$ for 90 min to remove the insoluble fraction, and the supernatant was loaded onto Anti-Flag G1 resin. The resin was washed with 20 column volume of wash buffer 1 (50 mM HEPES pH 7.5, 150 mM NaCl, 0.01% GDN, 0.001% CHS, 0.001% LMNG), then wash buffer 2 (20 column volume of 50 mM HEPES pH 7.5, 150 mM NaCl, 0.01% GDN, 0.001% CHS). And the target protein was eluted with elution buffer (50 mM HEPES pH 7.5, 150 mM NaCl, 0.01% GDN, 0.001% CHS, 200 ng/μL peptide). Protein was concentrated using an Amicon Ultra (MWCO 100 K, Millipore) centrifugal device and then purified by Superose 6 Increase 10/300GL gel filtration chromatography (GE Healthcare) in buffer containing 50 mM HEPES pH 7.5, 150 mM NaCl, 0.005% GDN, 0.0005% CHS. The peak fractions were pooled and concentrated for EM analysis. ABCB10 ATPase dead mutant E659Q was purified similarly. For cryo-EM study, protein was concentrated to ~4 mg/mL and incubated with 1 mM biliverdin (ChemeGen) for 30 min.

### ATP hydrolysis measurements

The ATPase activity was measured using QuantiChrom™ ATPase/GTPase assay kit (BioAssay Systems). The protein concentrations for the assays ranged from 0 to 1 μM. Reactions were performed using the reaction buffer from the assay kit plus 1 mM DTT and 1 mM ATP. Reactions were carried out at room temperature for 30 min and terminated by addition of the reagent from assay kit. The mixture was incubated for 30 min at room temperature before the activity was measured by monitoring the absorbance at 620 nm. To measure the ATPase activity of ABCB10 in the presence of cholesterol, cholesterol−methyl-β-cyclodextrin (sigma-aldrich catalog number: C4951-30MG) was dissolved in aqueous solution and the final concentration of cholesterol is 1.5 μg/mL unless otherwise mentioned. For comparison of the ATPase activity of different constructs of ABCB10 in the presence or absence of BV or cholesterol, a single ATP concentration (2 mM) was used and reactions were carried out at room temperature for 60 min.

The ABCB10 proteins in the CHS-supplemented DDM micelles for thermal stability measurements were purified similarly, except that GDN was replaced by 0.02% DDM plus 0.002% CHS in the purification procedures. The differential scanning fluorimetry (DSF) method was used to follow the thermal unfolding event of ABCB10 with a Prometheus NT.48 device (NanoTemper Technologies, Munich, Germany). Purified ABCB10 was diluted to 0.5 mg/mL and supplemented with decreasing amounts of biliverdin in a dilution series of 6 points starting at 50 μM down to 1 μM. The fluorescence at 330 and 350 nm was recorded over a temperature gradient scan from 25 °C to 95 °C and processed in GraphPad Prism 9.0 (GraphPad Software).

### Cryo-EM sample preparation

In all, 3.5 μL sample was applied onto a freshly glow discharged (70 s, 5 W, Plasma Cleaner Glow Discharge) 300 mesh R 1.2/1.3 holey carbon Film (Quantifoil) Cu grid[30]. Excess sample was blotted away using Whatman No. 1 filter paper with blot force 4, blot time 3−6 s and wait time 5 s then vitrified by plunge freezing into liquid ethane cooled by liquid nitrogen using vitrobot Mark IV (Thermo Fisher Scientific, USA).

### Cryo-EM data collection

Grids were clipped and loaded onto 300 kV Titan Krios electron microscope (Thermo Fisher) with autoloader. Images were recorded using Serial EM software and raw movies were collected at a nominal magnification of 105,000x (pixel size 0.85 Å) using K3 camera. Inelastically scattered electrons were excluded by a GIF Quantum energy filter (Gatan, USA) using a slit width of 20 eV. The movies were acquired with the defocus range of −1.1 to −1.8 μm with total exposure time of 2.5 s fragmented into 50 frames. The dose rate was 14.81 e-/(pix·s) and total dose was 55.21 e-/Å². EPU was used for semi-automatic data acquisition[31].

### Cryo-EM data processing, model building, and refinement

CryoSPARC V3.3.2 software package was used for data processing[32]. Patch Motion Correction was used to correct beam-induced motion correction and Patch CTF Estimation was used to calculate CTF parameters such as defocus and CTF fit resolution. Junk images were discarded by screening parameters such as CTF fit resolution, ice thickness and motion distance. Particles were auto-picked using template from PDB 4AYX and then subjected to 2D classification. Selected particles with an appropriate 2D average from 2D classification were further subjected to Ab-initio to classify the particles into four classes. Heterogeneous refinement was performed using the initial models generated from Ab-initio. Particles with high-resolution 3D average were selected and sent to Non-uniform refinement separately. C2 symmetry, CTF refinement and local refinement were performed to generate the final maps. PDB 4AYX was used as initial atomic model. Coot[33] and CCP-EM[34] were used for model building, refinement, and validation.

### Reporting summary

Further information on research design is available in the Nature Portfolio Reporting Summary linked to this article.

## Data availability

The data that support this study are available in a publicly accessible repository. The cryo-EM maps have been deposited in the Electron Microscopy Data Bank (EMDB) under accession codes EMD-33604 (ABCB10-apo), and EMD-33603 (ABCB10-BV). The coordinated have been deposited in the Protein Data Bank (PDB) under accession codes 7Y49 (ABCB10-apo), and 7Y48 (ABCB10-BV). Source data are provided with this paper. Other PDB entries used in this study: 4AYX, 3ZDQ, 4AYW. UniProtKB entries used in this study: Q9NRK6 (human ABCB10), Q9JI39 (mouse ABCB10), F1Q5K6 (zebrafish ABCB10), Q8SWW9 (Drosophila CG3156), P33310 (yeast multidrug resistance-like 1). Source data are provided with this paper.

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

## Acknowledgements

This work was supported by National Key Research and Development Program of China 2018YFE0204500 (Z.O.) and National Natural Science Foundation of China 31870751 (Z.O.).

## Author contributions

S.C.,Y.Y., X.Y., and H.S. devised the methodology; S.C., Y.Y., L.H., Y.H., X.Y., and H.S. performed the investigation; J.W. and Z.O. acquired the funding and supervised the project; S.C., Y.Y., L.H., J.W., and Z.O. wrote the paper with input from all authors.

## Competing interests

The authors declare no competing interests.
