## [Peer Review File · Nature Communications]

Cryo-EM Structures of mitochondrial ABC transporter ABCB10 in apo and biliverdin-bound formReviewers' Comments:

Reviewer #1:

Remarks to the Author:

The study by Cao, Yang et al. shows the Cryo-EM structure of human ABCB10, revealing novel and highly relevant structural states and properties of this mitochondrial biliverdin exporter. This study shows that: 1) Biliverdin delays thermal dissociation of purified ABCB10, validating the direct binding of biliverdin to ABCB10 as expected from the previously published transport studies. 2) Biliverdin can be visualized in the substrate binding site predicted by Shintre et al 2013, suggesting that one (N/I)xxR of the two signature motifs of ABCB10 directly participates in forming bonds with biliverdin (N229 with the carboxyl group of biliverdin). 3) ABCB10 bound to substrate (biliverdin) is still in inward-facing and closed conformation, with apo-ABCB10 being more open but still inward facing. These new data sets, together with the data from Shintre et al, support that both biliverdin and ATP might need to be simultaneously bound to ABCB10 to switch to an onward-facing conformation. 4) Two molecules of biliverdin are bound per ABCB10 dimers, setting the transport ratio biliverdin:ATP 1:1. Therefore, this study represents a major advancement in understanding how ABCB10 transports biliverdin and is the first study to visualize a substrate bound to ABCB10. However, some additional experiments are needed to confirm that the area where biliverdin is visualized to bind is indeed the site from where biliverdin is released when it is transported:

Major concerns:

1) Generate at least one ABCB10 mutant that would be expected to disrupt biliverdin binding and measure the ability of this mutant to:

1a) Increase ATPase activity in the presence of biliverdin.

1b) The ability of biliverdin to change the thermal stability of this mutant.

I would choose to mutate N229, as it is a highly conserved and signature residue of ABCB10. It is expected to form the key bond with the carboxylic group of biliverdin that would bring TMH6 closer and possibly complete the switch to an onward conformation when ATP is bound.

2) The conclusion that cholesterol is co-exported with biliverdin by ABCB10 is highly speculative and not supported by data in the literature. Indeed, ABCB10 deletion improves mitochondrial function in mice with high lipid and cholesterol levels. If ABCB10 was exporting cholesterol, ABCB10 KO liver mitochondria would be damaged in obese mice. To conclude that cholesterol co-export can be occurring, authors must at least measure the effects of free cholesterol on i) ABCB10 ATPase activity, ii) the thermal stability of ABCB10 and iii) whether an ABCB10 mutant that disrupts cholesterol and biliverdin binding has an effect on cholesterol, biliverdin or cholesterol+biliverdin expected ability to increase ATPase activity and ABCB10 thermal stability. Otherwise, this conclusion must be removed.

3) The authors conclude that: "The primary binding partner of cholesterol is biliverdin". There is no direct evidence to support this statement presented in the paper nor in the literature. It is a possibility that cholesterol is located there because that region of ABCB10 provides an environment that accommodates the interaction with hydrophobic and amphipatic molecules. Accordingly, cardiolipin was observed in this same region of ABCB10 in the study of Shintre et al and cardiolipin is not transported by ABCB10, as it could not activate ABCB10 ATPase activity. Authors must remove the conclusion of biliverdin being an interaction partner of cholesterol.

4) What is the source of the cholesterol detected in the structure coming from? Is it a product of hydrolysis of the cholesteryl hemisuccinate (CHS) used to solubilize ABCB10 or a contaminant of the biliverdin preparation? Why cholesterol is not visualized in apo-ABCB10 by Cryo-EM, when other lipids (cardiolipin) could be observed in the substrate binding site in ABCB10 crystal structure? These questions should be resolved and discussed.

Minor concerns

- 1) From the structure, one can see that it is biliverdin IXalpha not biliverdin IX beta, but it should be explicitly stated in the text.
- 2) It would be nice to test the effects of bilirubin and biliverdin IX beta on ABCB10 ATPase activity or thermal stability.
- 3) The manuscript should be revised for grammatical and typographical errors. There were ambiguous statements.

Reviewer #2:

Remarks to the Author:

In this manuscript Cao et. al present cryo-EM structures of the mitochondrial ABC transporter ABCB10 in apo and biliverdin (BV) bound forms. By capturing ABCB10 in a BV bound state the authors provide a critical piece of missing information regarding substrate transport by the ABCB10 transporter. Comparison of their structures with previously determined crystal structures of ABCB10 provide insight into the conformational changes the transporter undergoes during substrate capture.

While the cryo-EM data presented by the authors appears to be of high quality, I do not consider the manuscript in its current form to present an advance significant enough to warrant publication in Nature Communications. The justification for this position is explained below, along with suggestions to transform the manuscript into a higher quality form that could warrant publication.

Throughout the manuscript the authors focus heavily on the conformational changes of ABCB10 going from the apo to BV bound form. While these subtle conformational changes are interesting, they are not significantly different than what has been observed in many other ABC transporters. The recent explosion of ABC transporter cryo-EM structures has now painted a detailed picture of the inherent flexibility and conformational changes such proteins undergo during substrate capture and transport. This situation is particularly true for type I ABC exporters such as ABCB10. As such, I do not feel that a detailed analysis of these subtle conformational changes warrants a significant advance over what has already been demonstrated with other ABC transporters in recent literature.

In this reviewer's opinion, the major advance in this manuscript is the binding mode of BV within the transporter, and the finding that cholesterol also binds with BV within the substrate translocation pathway. The suggestion that cholesterol may be co-transported along with BV is a major advance, and should be explored in more depth throughout the manuscript. The authors should consider rearranging the manuscript significantly to strongly highlight the cholesterol binding, and potential co-transport mechanism. Such a co-transport mechanism represents a significant advance over what is known for similar ABC transporters, and would be of general interest to a much wider audience. If the authors could find a way to demonstrate through biochemical assays or other means that cholesterol indeed plays a large role in BV transport, or is even co-transported, the manuscript would be strengthened immensely.

In addition to the critique provided above, there are also several pieces of data analysis and presentation that the authors should clarify....

1. As mentioned above, one of the major findings in this paper is that cholesterol binds along with BV. However, there is not a single figure showing the cryo-EM map for cholesterol. At the very least the authors should show the cryo-EM map for this bound co-substrate, and consider dedicating an entire main figure to describing cholesterol interactions. I would suggest moving figure 2 to the supplement, and using this main figure space to more clearly describe cholesterol interactions.

2. The authors suggest that cholesterol is bound in the translocation pathway along with BV. However, a significant amount of cholesteryl-hemisuccinate (CHS) was included throughout membrane solubilization and purification. Why do the authors believe that they have captured cholesterol and not CHS in the transport pathway? Can this distinction be clearly made from the cryo-EM map alone? Is it possible that the "cholesterol" observed in the cryo-EM map is simply an artifact of the detergent system used for purification and structure determination?

3. As described above, it seems likely that the "cholesterol" observed in the cryo-EM map could be CHS. Shum, M. et. al previously reconstituted ABCB10 in liposomes that lack cholesterol and found that ABCB10 could still transport BV. Do the authors believe that the "cholesterol" they observe in their cryo-EM map is a strict requirement for BV transport? The authors should consider performing some type of validation to see if cholesterol has any effect on BV transport or ATPase activity. This could be conducted in the form of a BV transport assay (or at a minimum ATPase stimulation) in liposomes with and without cholesterol. I realize that such assays are not trivial and represent a significant amount of extra experimentation, but in this particular case I feel such additional studies are required to significantly strengthen the claims made in the manuscript.

4. At the end of the first paragraph of the results section the authors describe results of thermofluor TM shift. The melting temperatures described in Figure S1C and S1D vary substantially from one another. In figure S1C ABCB10-apo has a T_m of ~49.3 degrees, and in figure S1D the ABCB10-apo has a T_m of ~45 degrees. This discrepancy in T_m for ABCB10-apo between figure panel S1C and S1D is ~4.5 degrees, a highly significant difference in T_m, and a difference that is much greater than the reported T_m shift induced by BV itself. How do the authors explain this significant discrepancy?

5. Along the lines of point 4 above, why was GDN-CHS exchanged for DDM-CHS in the thermofluor assays? Similarly, the Prometheus NT.48 device measures fluorescence at 330 and 350nm, wavelengths that are strongly absorbed by biliverdin. Did the authors find that using biliverdin with this device resulted in any abnormalities in data recording or analysis?

6. The authors mention that the "secondary structures of ABCB10-BV are more well defined than that of ABCB10-apo". In the methods section the authors only mention using COOT and CCP-EM for model building and refinement. What type of restraints were used during atomic model refinement using REFMAC? Was it only the default jelly-body restraints, or were reference model restraints included as well? Were secondary structure restraints used during atomic model refinement? In figure 3 it appears that TM helices and the majority of the beta-sheets in the NBD do not have properly assigned secondary structure.

7. Along with point 5 above, in figure 4b the authors suggest that the distance between ATP binding sites is unequal in ABCB10-apo as a "result of being slightly twisted". If C₂ symmetry was applied during map reconstruction of ABCB10-apo, then I don't see how there could possibly be unequal distance between the ATP binding sites. The map should be a perfect 180 degree rotation upon itself, and thus the distances between ATP binding site should be equal. I would suggest that the "unequal distance" that is observed is simply an artifact of refining the model into weak NBD density without proper restraints to avoid model distortion.

8. The authors mention that "We therefore believe that the ATPase activity of ABCB10 can increase dramatically in the presence of BV." Indeed, this has already been demonstrated by Shum, M. et al. with liposome reconstituted ABCB10, and the authors should cite this previous work after this statement in the manuscript.

9. In Figure S4 the authors show that one of the carboxylates of BV is located next to an area of high positive charge. What residue(s) in ABCB10 is creating this local area of positive charge? Similarly, are there any point mutants that the authors can make to abolish BV binding or transport? Are there any known mutations in ABCB10 that cause disease in humans, particularly with respect to the BV binding

pocket?

10. It has previously been demonstrated that biliverdin but not bilirubin can stimulate the ATPase activity of ABCB10 (Shum, M. et al.). Since biliverdin and bilirubin are very similar in overall structure, differing only in an additional hydrogen and slight double-bond rearrangement, can the authors speculate based on their cryo-EM structures why ABCB10 is specific for biliverdin?

11. The citations in the manuscript are given as numbers, but the bibliography is in alphabetical order with no numbers. Thus, I have no possible way of knowing if citations point to proper references throughout the manuscript.

12. Lastly, please check proper spelling and grammar use throughout the manuscript. For instance, the word "serve" is improperly written as "sever" throughout the manuscript. Without line numbers in the manuscript it is far too laborious as a reviewer to point out all of these such errors. In the future, please consider including line numbers in your initial manuscript submission, as it greatly helps reviewers to point out specific areas that require attention.

Review 1:

The study by Cao, Yang et al. shows the Cryo-EM structure of human ABCB10, revealing novel and highly relevant structural states and properties of this mitochondrial biliverdin exporter. This study shows that:

- 1) Biliverdin delays thermal diassociation of purified ABCB10, validating the direct binding of biliverdin to ABCB10 as expected from the previously published transport studies.*
- 2) Biliverdin can be visualized in the substrate binding site predicted by Shintre et al 2013, suggesting that one (N/I)xxR of the two signature motifs of ABCB10 directly participates in forming bonds with biliverdin (N229 with the carboxyl group of biliverdin).*
- 3) ABCB10 bound to substrate (biliverdin) is still in inward-facing and closed confirmation, with apo-ABCB10 being more open but still inward facing. These new data sets, together with the data from Shintre et al, support that both biliverdin and ATP might need to be simultaneously bound to ABCB10 to switch to an onward-facing conformation.*
- 4) Two molecules of biliverdin are bound per ABCB10 dimers, setting the transport ratio biliverdin:ATP 1:1. Therefore, this study represents a major advancement in understanding how ABCB10 transports biliverdin and is the first study to visualize a substrate bound to ABCB10. However, some additional experiments are needed to confirm that the area where biliverdin is visualized to bind is indeed the site from where biliverdin is released when it is transported:*

Response: We thank the reviewer for the positive comments.

1. Major concerns:

1) Generate at least one ABCB10 mutant that would be expected to disrupt biliverdin binding and measure the ability of this mutant to:

1a) Increase ATPase activity in the presence of biliverdin.

1b) The ability of biliverdin to change the thermal stability of this mutant.

I would choose to mutate N229, as it is a highly conserved and signature residue of ABCB10. It is expected to form the key bond with the carboxylic group of biliverdin that would bring TMH6 closer and possibly complete the switch to an onward conformation when ATP is bound.

Response: We thank the reviewer for these good suggestions. We made several ABCB10 mutants, including N229A, F398A, N407A, N229A/F398A, N229A/N407A, and N229A/F398A/N407A, and tested their ATPase activities in the presence or absence of biliverdin and their abilities of biliverdin binding through thermo shift assays. The results show that biliverdin can stimulate the ATPase activity of ABCB10 WT but not ABCB10 mutants except N229A. Consistently, BV increases the thermal stability of ABCB10 WT much significantly than that of ABCB10 mutants (Figure 4E-F). These results validate the biliverdin binding sites in our cryo-EM structure. These new results are incorporated into the revised manuscript.

2) The conclusion that cholesterol is co-exported with biliverdin by ABCB10 is highly

speculative and not supported by data in the literature. Indeed, ABCB10 deletion improves mitochondrial function in mice with high lipid and cholesterol levels. If ABCB10 was exporting cholesterol, ABCB10 KO liver mitochondria would be damaged in obese mice. To conclude that cholesterol co-export can be occurring, authors must at least measure the effects of free cholesterol on i) ABCB10 ATPase activity, ii) the thermal stability of ABCB10 and iii) whether an ABCB10 mutant that disrupts cholesterol and biliverdin binding has an effect on cholesterol, biliverdin or cholesterol+biliverdin expected ability to increase ATPase activity and ABCB10 thermal stability. Otherwise, this conclusion must be removed.

Response: We agree with the reviewer and it's really critical comments. We measured the T_m of ABCB10 WT and mutants in the presence of cholesterol. We measured the ATPase activity and T_m of ABCB10 in the presence or absence of cholesterol. Cholesterol promotes the ATPase activity of ABCB10 in a dose-dependent manner. The ability of cholesterol to stimulate the ATPase activity of ABCB10 mutants such as N229A, F398A, N407A and N229A/F398A is compromised or abolished (Fig.5B-C). Both cholesterol and BV enhance the thermal stability of ABCB10 WT or N229A. Incubation with two compounds simultaneously displays synergic effect Therefore, cholesterol is cofactor of ABCB10. Even so, it is premature to conclude that cholesterol is co-exported with BV. We move this highly speculative hypothesis to discussion part.

3) The authors conclude that: "The primary binding partner of cholesterol is biliverdin". There is no direct evidence to support this statement presented in the paper nor in the literature. It is a possibility that cholesterol is located there because that region of ABCB10 provides an environment that accomodates the interaction with hydrophobic and amphipatic molecules. Accordingly, cardiolipin was observed in this same region of ABCB10 in the study of Shintre et al and cardiolipin is not transported by ABCB10, as it could not activate ABCB10 ATPase activity. Authors must remove the conclusion of biliverdin being an interaction partner of cholestherol.

Response: We thank the reviewer for this good suggest. In our structure two cholesterols are sandwiched by two BV molecules. We agree with the reviewer that it is possible that cholesterols are located there because that region of ABCB10 provides an environment that accommodates the interaction with hydrophobic and amphipatic molecules. We have modified it in our revised manuscript.

4) What is the source of the cholesterol detected in the structure coming from? Is it a product of hydrolysis of the cholesteryl hemisuccinate (CHS) used to solubilize ABCB10 or a contaminant of the biliverdin preparation? Why cholesterol is not visualized in apo-ABCB10 by Cryo-EM, when other lipids (cardiolipin) could be observed in the substrate binding site in ABCB10 crystal structure? These questions should be resolved and discussed.

Response: We thank the reviewer for this critical question. Since CHS is a derivative of cholesterol, it is possible that cholesterol is a product of hydrolysis of CHS. It may also come from contaminant of biliverdin or other reagents. However, cholesterol might only bind there when biliverdin occupies the translocation pathway and provides an environment that favors

the interaction.

Minor concerns

1) *From the structure, one can see that it is biliverdin IXalpha not biliverdin IX beta, but it should be explicitly stated in the text.*

Response: We thank the reviewer for this great suggestion. We have clarified it in the text.

2) *It would be nice to test the effects of bilirubin and biliverdin IX beta on ABCB10 ATPase activity or thermal stability.*

Response: It has previously been demonstrated that biliverdin but not bilirubin can stimulate the ATPase activity of ABCB10 reconstituted into nanodiscs (Shum, M. et al. 2021). In our experimental settings, however, we found that bilirubin can stimulate the ATPase activity of pure ABCB10 in solution, but not as significantly as biliverdin does (data not shown). We did not test the effect of biliverdin IX beta on ATPase activity as this compound is not available in the market.

3) *The manuscript should be revised for grammatical and typographical errors. There were ambiguous statements.*

Response: We did English proofreading to minimize the language errors.

Reviewer #2 (Remarks to the Author):

In this manuscript Cao et. al present cryo-EM structures of the mitochondrial ABC transporter ABC10 in apo and biliverdin (BV) bound forms. By capturing ABCB10 in a BV bound state the authors provide a critical piece of missing information regarding substrate transport by the ABCB10 transporter. Comparison of their structures with previously determined crystal structures of ABCB10 provide insight into the conformational changes the transporter undergoes during substrate capture.

While the cryo-EM data presented by the authors appears to be of high quality, I do not consider the manuscript in its current form to present an advance significant enough to warrant publication in Nature Communications. The justification for this position is explained below, along with suggestions to transform the manuscript into a higher quality form that could warrant publication.

Throughout the manuscript the authors focus heavily on the conformational changes of ABCB10 going from the apo to BV bound form. While these subtle conformational changes are interesting, they are not significantly different than what has been observed in many other

ABC transporters. The recent explosion of ABC transporter cryo-EM structures has now painted a detailed picture of the inherent flexibility and conformational changes such proteins undergo during substrate capture and transport. This situation is particularly true for type I ABC exporters such as ABCB10. As such, I do not feel that a detailed analysis of these subtle conformational changes warrants a significant advance over what has already been demonstrated with other ABC transporters in recent literature.

In this reviewer's opinion, the major advance in this manuscript is the binding mode of BV within the transporter, and the finding that cholesterol also binds with BV within the substrate translocation pathway. The suggestion that cholesterol may be co-transported along with BV is a major advance, and should be explored in more depth throughout the manuscript. The authors should consider rearranging the manuscript significantly to strongly highlight the cholesterol binding, and potential co-transport mechanism. Such a co-transport mechanism represents a significant advance over what is known for similar ABC transporters, and would be of general interest to a much wider audience. If the authors could find a way to demonstrate through biochemical assays or other means that cholesterol indeed plays a large role in BV transport, or is even co-transported, the manuscript would be strengthened immensely.

Response: We thank the reviewer for these great suggestions, we have rearranged the manuscript significantly according to the reviewer's suggestion. We highlighted the BV binding and validated the BV-binding surface of ABCB10 by measuring the ATPase activity and thermal stability of ABCB10 WT and ABCB10 mutants. We also included a main figure for cholesterol interaction and its role in ATPase activity and thermal stability.

In addition to the critique provided above, there are also several pieces of data analysis and presentation that the authors should clarify....

1. As mentioned above, one of the major findings in this paper is that cholesterol binds along with BV. However, there is not a single figure showing the cryo-EM map for cholesterol. At the very least the authors should show the cryo-EM map for this bound co-substrate, and consider dedicating an entire main figure to describing cholesterol interactions. I would suggest moving figure 2 to the supplement, and using this main figure space to more clearly describe cholesterol interactions.

Response: We thank the reviewer for this great suggestion. We have a main figure showing the cholesterol binding, its role in ATPase activity and thermal stability in our revised manuscript (Figure 5). We also show the cryo-EM map for cholesterol (Supplementary Fig. 4C). We have moved figure 2 to the supplement according to the reviewer's suggestion. We think that our manuscript has improved a lot after revision.

2. The authors suggest that cholesterol is bound in the translocation pathway along with BV. However, a significant amount of cholesteryl-hemisuccinate (CHS) was included throughout membrane solubilization and purification. Why do the authors believe that they have captured cholesterol and not CHS in the transport pathway? Can this distinction be clearly

made from the cryo-EM map alone? Is it possible that the “cholesterol” observed in the cryo-EM map is simply an artifact of the detergent system used for purification and structure determination?

Response: It is true that cholesteryl-hemisuccinate (CHS) has similar structure with cholesterol and it's hard to tell whether it is CHS or cholesterol only from electron densities. We measured the T_m and ATPase activity of ABCB10 WT and mutants in the presence of cholesterol. Cholesterol enhances the ATPase activity of ABCB10 WT in a dose-dependent manner. The ability of cholesterol to stimulate the ATPase activity of ABCB10 mutants such as N229A, F398A, N407A and N229A/F398A is compromised or abolished (Fig.5B-C). Both cholesterol and BV enhance the thermal stability of ABCB10 WT or N229A. Incubation with two compounds simultaneously displays synergic effect (Fig.5D). The rationale is that these mutants compromise or abolish the interaction with BV, while BV binding with ABCB10 is a prerequisite for cholesterol interaction. Therefore, we believe that this density is cholesterol not CHS. We have included this result in the revised manuscript.

3. As described above, it seems likely that the “cholesterol” observed in the cryo-EM map could be CHS. Shum, M. et. al previously reconstituted ABCB10 in liposomes that lack cholesterol and found that ABCB10 could still transport BV. Do the authors believe that the “cholesterol” they observe in their cryo-EM map is a strict requirement for BV transport? The authors should consider performing some type of validation to see if cholesterol has any effect on BV transport or ATPase activity. This could be conducted in the form of a BV transport assay (or at a minimum ATPase stimulation) in liposomes with and without cholesterol. I realize that such assays are not trivial and represent a significant amount of extra experimentation, but in this particular case I feel such additional studies are required to significantly strengthen the claims made in the manuscript.

Response: As we mentioned above, we measured the T_m and ATPase activity of ABCB10 WT and mutants in the presence of cholesterol. Cholesterol enhances the ATPase activity of ABCB10 WT in a dose-dependent manner. The ability of cholesterol to stimulate the ATPase activity of ABCB10 mutants such as N229A, F398A, N407A and N229A/F398A is compromised or abolished (Fig.5B-C). Both cholesterol and BV enhance the thermal stability of ABCB10 WT or N229A. Incubation with two compounds simultaneously displays synergic effect (Fig.5D). We therefore believe that the density is cholesterol instead of CHS. Therefore, cholesterol is cofactor of ABCB10. Even so, it is premature to conclude that cholesterol is co-exported with BV. We move this highly speculative hypothesis to discussion part.

4. At the end of the first paragraph of the results section the authors describe results of thermofluor T_m shift. The melting temperatures described in Figure S1C and S1D vary substantially from one another. In figure S1C ABCB10-apo has a T_m of ~49.3 degrees, and in figure S1D the ABCB10-apo has a T_m of ~45 degrees. This discrepancy in T_m for ABCB10-apo between figure panel S1C and S1D is ~4.5 degrees, a highly significant difference in T_m, and a difference that is much greater than the reported T_m shift induced by BV itself. How do the authors explain this significant discrepancy?

Response: We thank the reviewer for the comments and we are sorry about the confusion. The protein concentrations were different in these two experiments. In Figure S1C, the ABCB10 protein concentration is 0.3 mg/ml while in Figure S1D it is 0.5 mg/ml. We repeated the experiment in Figure S1C with 0.5 mg/ml of ABCB10. The measured T_m is 44.85 degree which is similar with that in Figure S1D (45.1 degree). We have updated the results in our revised version.

5. *Along the lines of point 4 above, why was GDN-CHS exchanged for DDM-CHS in the thermofluor assays? Similarly, the Prometheus NT.48 device measures fluorescence at 330 and 350nm, wavelengths that are strongly absorbed by biliverdin. Did the authors find that using biliverdin with this device resulted in any abnormalities in data recording or analysis?*

Response: We did not exchange detergent intentionally, just because we run out of GDN during the experiment and found out that DDM-CHS also works. We measured the absorption of BV alone without any protein (100 μ M biliverdin) and did not observe strong absorption.

6. *The authors mention that the “secondary structures of ABCB10-BV are more well defined than that of ABCB10-apo”. In the methods section the authors only mention using COOT and CCP-EM for model building and refinement. What type of restraints were used during atomic model refinement using REFMAC? Was it only the default jelly-body restraints, or were reference model restraints included as well? Were secondary structure restraints used during atomic model refinement? In figure 3 it appears that TM helices and the majority of the beta-sheets in the NBD do not have properly assigned secondary structure.*

Response: We included secondary structure restraints, rotamer restraints, ramachandran restraints and NCS restraints during model refinement using Phenix. Due to the low local resolution of the NBD in ABCB10-apo structure (Supplementary Figure 3: Cryo-EM maps and refined structures of ABCB10-apo), TM helices and the majority of the beta-sheets in the NBD do not have properly assigned secondary structure. The resolution of the corresponding part in ABCB10-BV structure is high enough to build the secondary structures.

7. *Along with point 5 above, in figure 4b the authors suggest that the distance between ATP binding sites is unequal in ABCB10-apo as a “result of being slightly twisted”. If C2 symmetry was applied during map reconstruction of ABCB10-apo, then I don’t see how there could possibly be unequal distance between the ATP binding sites. The map should be a perfect 180 degree rotation upon itself, and thus the distances between ATP binding site should be equal. I would suggest that the “unequal distance” that is observed is simply an artifact of refining the model into weak NBD density without proper restraints to avoid model distortion.*

Response: We thank the reviewer for the comments. Yes, we applied C2 symmetry in both ABCB10-apo and ABCB10-BV, the distance should be equal and the difference is caused by model refinement and distortions.

8. *The authors mention that “We therefore believe that the ATPase activity of ABCB10 can increase dramatically in the presence of BV.” Indeed, this has already been demonstrated by Shum, M. et al. with liposome reconstituted ABCB10, and the authors should cite this previous work after this statement in the manuscript.*

Response: We thank the reviewer for this suggestion. We have modified the text and added the citation accordingly.

9. *In Figure S4 the authors show that one of the carboxylates of BV is located next to an area of high positive charge. What residue(s) in ABCB10 is creating this local area of positive charge? Similarly, are there any point mutants that the authors can make to abolish BV binding or transport? Are there any known mutations in ABCB10 that cause disease in humans, particularly with respect to the BV binding pocket?*

Response: We thank the reviewer for the question. Residues R295 and R232 in ABCB10 create the local area of positive charge. We made several ABCB10 mutants, including N229A, F398A, N407A, N229A/F398A, N229A/N407A, and N229A/F398A/N407A, and tested their ATPase activities and abilities of biliverdin binding through thermo shift assays. The results show that BV stimulates the ATPase of ABCB10 WT, but not the mutants except N229A. Consistently, BV increases the thermal stability of ABCB10 WT much significantly than that of ABCB10 mutants (Figure 4E-F). These results are incorporated into the revised manuscript. To the best of our knowledge, we did not find any report of aforementioned mutations in ABCB10 that cause disease in human.

10. *It has previously been demonstrated that biliverdin but not bilirubin can stimulate the ATPase activity of ABCB10 (Shum, M. et al.). Since biliverdin and bilirubin are very similar in overall structure, differing only in an additional hydrogen and slight double-bond rearrangement, can the authors speculate based on their cryo-EM structures why ABCB10 is specific for biliverdin?*

Response: This is a tough but fundamental question. We cannot distinguish biliverdin from bilirubin based on our cryo-EM map. We cannot distinguish them based on the ligand binding pattern or conformation either in our structure. We tested the ATPase activity of ABCB10 in the presence of bilirubin or biliverdin. we found that bilirubin can stimulate the ATPase activity of pure ABCB10 in solution, but not as significantly as biliverdin does (data not shown). It takes several conformational changes to achieve BV export, one conformation of ABCB10, especially in the initial stage of the process, may prefer BV than bilirubin.

11. The citations in the manuscript are given as numbers, but the bibliography is in alphabetical order with no numbers. Thus, I have no possible way of knowing if citations point to proper references throughout the manuscript.

Response: We feel sorry about the confusing citation style. We have formatted citations according to the Journal's formatting instructions.

12. Lastly, please check proper spelling and grammar use throughout the manuscript. For instance, the word "serve" is improperly written as "sever" throughout the manuscript. Without line numbers in the manuscript it is far too laborious as a reviewer to point out all of these such errors. In the future, please consider including line numbers in your initial manuscript submission, as it greatly helps reviewers to point out specific areas that require attention.

Response: We thank the review for the correction. We have checked the spelling and grammar use throughout the manuscript and corrected the improper writing. We also added line numbers for each page in the manuscript.

In summary, we have addressed all comments from the reviewers. We feel that our manuscript has improved a lot after revision. We hope that you will find our revised manuscript now suitable for publication.

Sincerely,

Zhuqing Ouyang, Ph.D

Reviewers' Comments:

Reviewer #1:

Remarks to the Author:

The authors successfully addressed the major concerns raised during the previous round of revision. Indeed, the data with multiple mutants goes beyond what was requested. These findings are highly significant for the field, providing unprecedented knowledge on ABCB10 function and thus warranting publication in a journal such as Nature Communications. However, there are some minor concerns regarding data interpretation around the role of cholesterol binding in ABCB10 function that need to be changed:

1) Authors conclude that "Both cholesterol and BV enhance the thermal stability of ABCB10 WT or N229A. Incubation with two compounds simultaneously displays synergic effect (Fig.5D)." However, cholesterol by itself decreases the T_m of ABCB10 and thus thermal stability (Figure 5B), which supports that the actions of cholesterol increasing ATP hydrolysis are mechanistically different than those of biliverdin.

2) The fact that cholesterol by itself induces a small decrease in T_m while concurrently increasing ATPase activity, suggests that cholesterol is modulating basal ATPase activity without playing a direct role in BV binding or transport. Figure 5D shows that biliverdin limits the ability of cholesterol to increase ATPase activity. Thus, it can only be concluded that cholesterol regulates ABCB10 ATPase activity independently of biliverdin transport. The conclusion of being a co-factor needed for transport should be removed, as indeed Shum et al. already showed biliverdin transport without cholesterol.

3) It would be nice to propose a hypothesis to reconcile the fact that cholesterol cannot be seen in ABCB10-Apo, despite cholesterol can stimulate ATPase activity. The divergence between stimulation of ATPase activity and the effect on T_m supports that cholesterol transiently binds to other regions of ABCB10 to stimulate ATPase activity, with this interaction decreasing T_m to favor basal ATPase activity. Maybe cholesterol binds with low affinity to the ATPase binding domain to induce a rapid conformational change, which could explain the poor resolution of the ATP binding domain in the apo form? This could explain why biliverdin decreased the efficacy of cholesterol-induced ATPase activity.

Reviewer #2:

Remarks to the Author:

Cao et al. have significantly rearranged their initial manuscript and provided further experiments to address the initial comments from reviewers. I appreciate the authors efforts to address all of the reviewers comments, and I do think their manuscript is significantly strengthened with this new data. While I think the suggestion from the authors that cholesterol is a co-transported substrate along with BV is still highly speculative, the added in-vitro data does seem to strengthen their overall argument that cholesterol may be a key player in the transport mechanism. However, the newly presented data also raise a few new questions that should be addressed....

1. In response to the original reviewer #2 comment about unequal distances between ATP binding sites in ABCB10-apo the authors responded "we applied C2 symmetry in both ABCB10-apo and ABCB10-BV, the distance should be equal and the difference is caused by model refinement and distortions". However, in the revised manuscript on page 8 line 18 the authors still state "the two NBDs of ABCB10-apo are slightly twisted relative to each other as revealed by the unequal distance between the two ATP binding sites". Similarly, figure 3B still shows an unequal distance between ATP binding sites, which is not possible if the cryo-EM map was determined with C2 symmetry and the model was correctly refined with NCS restraints.

2. The authors perform ATPase and thermofluor assays in the presence of cholesterol. Cholesterol is

notoriously difficult to solubilize in aqueous solutions, even in the presence of mild detergents. There is no mention of how the cholesterol was dissolved/prepared for subsequent in-vitro assays. The methods section should be amended to include how cholesterol was solubilized and added to the protein preparations for the in-vitro assays. Was an organic solvent used for this process?

3. The results of ATPase assays are reported in units of OD650 nm/nM. This is a very unusual way of reporting ATPase activity. Conversion to a more conventional unit of specific activity such as nmol ATP/min/mg protein would allow comparison with ATPase rates of other transporters in the literature.

4. Along the lines of point 3 above, as far as I can tell the concentration of ATP in all ATPase assays in the manuscript is 1mM. Why was 1mM ATP chosen, and how far above the K_m for the ATP reaction is this value? Usually when comparing ATPase activity of mutants a significantly higher ATP concentration would be chosen to ensure that V_{max} conditions are maintained. Can the authors be sure that the differences in ATPase activity observed with the mutants results from altered V_{max} as opposed to K_m ?

5. Based on the structure of ABCB10 bound to BV and cholesterol it is apparent that cholesterol makes extensive interactions with the bound BV. One would expect that in the absence of BV cholesterol may not bind (or bind very weakly) to ABCB10. This situation seems to be reflected in the result of Figure 5B where cholesterol alone has no effect on the TM of WT ABCB10. However, addition of cholesterol alone (without BV) was able to significantly increase the ATPase activity of WT ABCB10 (Fig 5D). How do the authors explain a lack of TM shift in the presence of cholesterol, but a massive increase in ATPase activity in the presence of this compound?

6. The authors claim that binding of cholesterol and BV is "synergistic". By definition synergism implies that the effect of adding cholesterol and BV to ABCB10 should produce an increased ATPase activity that is greater than that observed in the presence of cholesterol or BV alone. As seen in figure 5D this is clearly not the case, as cholesterol alone stimulated the ATPase activity to a greater extent than cholesterol + BV. Thus, based on the authors own data the effects of cholesterol and BV are NOT synergistic.

7. Page 12 line 20, the authors speculate that their structure of ABCB10 in the absence of BV may represent the "real" apo form. One could also easily make the argument that the conformations observed in this study may not be "real" as they report on the conformation of ABCB10 solubilized in detergent rather than a true lipid bilayer. While it is appropriate to describe differences between the apo conformation observed here, and the nucleotide free structure observed previously, I would strongly discourage the use of the word "real". In reality, the true "apo" form of ABCB10 may likely be represented by an ensemble of conformational states, which may contain both the observed "apo" and "nucleotide free" conformations.

8. Figure 5C caption mentions mutants...there are no mutants reported in this figure.

Reviewer #1 (Remarks to the Author):

The authors successfully addressed the major concerns raised during the previous round of revision. Indeed, the data with multiple mutants goes beyond what was requested. These findings are highly significant for the field, providing unprecedented knowledge on ABCB10 function and thus warranting publication in a journal such as Nature Communications. However, there are some minor concerns regarding data interpretation around the role of cholesterol binding in ABCB10 function that need to be changed:

Response: We thank the reviewer for the positive comments.

1) Authors conclude that "Both cholesterol and BV enhance the thermal stability of ABCB10 WT or N229A. Incubation with two compounds simultaneously displays synergic effect (Fig.5D)." However, cholesterol by itself decreases the T_m of ABCB10 and thus thermal stability (Figure 5B), which supports that the actions of cholesterol increasing ATP hydrolysis are mechanistically different than those of biliverdin.

Response: We thank the reviewer for these comments. Fig.5 was mistakenly appended to the manuscript and different from the source file uploaded separately. In our revised version, Fig.5d shows that both cholesterol and BV enhance the thermal stability of ABCB10 WT or N229A. Incubation with two compounds simultaneously displays synergic effect.

2) The fact that cholesterol by itself induces a small decrease in T_m while concurrently increasing ATPase activity, suggests that cholesterol is modulating basal ATPase activity without playing a direct role in BV binding or transport. Figure 5D shows that biliverdin limits the ability of cholesterol to increase ATPase activity. Thus, it can only be concluded that cholesterol regulates ABCB10 ATPase activity independently of biliverdin transport. The conclusion of being a co-factor needed for transport should be removed, as indeed Shum et al. already showed biliverdin transport without cholesterol.

Response: We thank the reviewer for the suggestion. As we mentioned above, Fig.5 was mistakenly appended to the manuscript and different from the source file uploaded separately. In our revised version, Fig.5d shows that both cholesterol and BV enhance the thermal stability of ABCB10 WT or N229A. Figure 5c shows that biliverdin limits the ability of cholesterol to increase ATPase activity. It might be possible that cholesterol regulates ABCB10 ATPase activity independently of biliverdin transport and the conclusion of being a co-factor needed for transport has been removed in our revised version.

3) It would be nice to propose a hypothesis to reconcile the fact that cholesterol cannot be seen in ABCB10-Apo, despite cholesterol can stimulate ATPase activity. The divergence between stimulation of ATPase activity and the effect on T_m supports that cholesterol

transiently binds to other regions of ABCB10 to stimulate ATPase activity, with this interaction decreasing T_m to favor basal ATPase activity. Maybe cholesterol binds with low affinity to the ATPase binding domain to induce a rapid conformational change, which could explain the poor resolution of the ATP binding domain in the apo form? This could explain why biliverdin decreased the efficacy of cholesterol-induced ATPase activity.

Response: We thank the reviewer for this good suggest. The reason that cholesterol can be seen in ABCB10-BV but not in ABCB10-Apo might be that that region of ABCB10-BV provides an environment that accommodates the interaction with hydrophobic and amphipatic molecules. However, cholesterol is too small to fit the pocket without BV in ABCB10-Apo structure. Both cholesterol and BV enhance the thermal stability of ABCB10, while biliverdin decreased the efficacy of cholesterol-induced ATPase activity. It might be possible that cholesterol regulates ABCB10 ATPase activity independently of biliverdin transport.

Reviewer #2 (Remarks to the Author):

Cao et al. have significantly rearranged their initial manuscript and provided further experiments to address the initial comments from reviewers. I appreciate the authors efforts to address all of the reviewers comments, and I do think their manuscript is significantly strengthened with this new data. While I think the suggestion from the authors that cholesterol is a co-transported substrate along with BV is still highly speculative, the added in-vitro data does seem to strengthen their overall argument that cholesterol may be a key player in the transport mechanism. However, the newly presented data also raise a few new questions that should be addressed....

Response: We thank the reviewer for the overall positive comments.

- 1. In response to the original reviewer #2 comment about unequal distances between ATP binding sites in ABCB10-apo the authors responded "we applied C2 symmetry in both ABCB10-apo and ABCB10-BV, the distance should be equal and the difference is caused by model refinement and distortions". However, in the revised manuscript on page 8 line 18 the authors still state "the two NBDs of ABCB10-apo are slightly twisted relative to each other as revealed by the unequal distance between the two ATP binding sites". Similarly, figure 3B still shows an unequal distance between ATP binding sites, which is not possible if the cryo-EM map was determined with C2 symmetry and the model was correctly refined with NCS restraints.*

Response: We thank the reviewer for this great suggestion. We have removed the statement and modified Figure 3b accordingly.

- 2. The authors perform ATPase and thermofluor assays in the presence of cholesterol. Cholesterol is notoriously difficult to solubilize in aqueous solutions, even in the presence of mild detergents. There is no mention of how the cholesterol was dissolved/prepared for subsequent in-vitro assays. The methods section should be amended to include how cholesterol was solubilized and added to the protein preparations for the in-vitro assays.*

Was an organic solvent used for this process?

Response: We thank the reviewer for reminding us. Indeed, we had problems in solubilizing cholesterol in aqueous solutions. It takes several tedious steps to prepare cholesterol–methyl- β -cyclodextrin (reference: Cholesterol Addition to ER Membranes Alters Conformation of SCAP, the SREBP Escort Protein that Regulates Cholesterol Metabolism, 2002). Later on, we found that sigma-aldrich sells this product as powder with catalog number: C4951-30MG. It solubilizes well in aqueous solution. We have added this information in the method section in our revised version.

3. *The results of ATPase assays are reported in units of OD650 nm/nM. This is a very unusual way of reporting ATPase activity. Conversion to a more conventional unit of specific activity such as nmol ATP/min/mg protein would allow comparison with ATPase rates of other transporters in the literature.*

Response: We thank the reviewer for the suggestion. We have converted the unit of ATPase activity at nmol ATP/min/mg protein.

4. *Along the lines of point 3 above, as far as I can tell the concentration of ATP in all ATPase assays in the manuscript is 1mM. Why was 1mM ATP chosen, and how far above the K_m for the ATP reaction is this value? Usually when comparing ATPase activity of mutants a significantly higher ATP concentration would be chosen to ensure that V_{max} conditions are maintained. Can the authors be sure that the differences in ATPase activity observed with the mutants results from altered V_{max} as opposed to K_m ?*

Response: We did ATPase assays in Supplementary Fig.1 with 1 mM of ATP and in Fig4-5 with 2 mM of ATP. We choose these concentrations because it is reported that K_m of ABCB10 in detergent is about 0.2 mM (Shintre et al. PNAS, 2013). Considering that they did the assays at 37°C and our experiment was carried out at room temperature, 2 mM is pretty safe ($[ATP] > 10 * K_m$) to maintain V_{max} conditions.

5. *Based on the structure of ABCB10 bound to BV and cholesterol it is apparent that cholesterol makes extensive interactions with the bound BV. One would expect that in the absence of BV cholesterol may not bind (or bind very weakly) to ABCB10. This situation seems to be reflected in the result of Figure 5B where cholesterol alone has no effect on the TM of WT ABCB10. However, addition of cholesterol alone (without BV) was able to significantly increase the ATPase activity of WT ABCB10 (Fig 5D). How do the authors explain a lack of TM shift in the presence of cholesterol, but a massive increase in ATPase activity in the presence of this compound?*

Response: Fig.5 was mistakenly appended to the manuscript and different from the source file uploaded separately. In our revised version, Fig.5d shows that both cholesterol and BV enhance the thermal stability of ABCB10 WT or N229A. Incubation with two compounds

simultaneously displays synergic effect. It is also possible that cholesterol transiently binds to other regions of ABCB10 to stimulate ATPase activity.

6. *The authors claim that binding of cholesterol and BV is "synergistic". By definition synergism implies that the effect of adding cholesterol and BV to ABCB10 should produce an increased ATPase activity that is greater than that observed in the presence of cholesterol or BV alone. As seen in figure 5D this is clearly not the case, as cholesterol alone stimulated the ATPase activity to a greater extent than cholesterol + BV. Thus, based on the authors own data the effects of cholesterol and BV are NOT synergistic.*

Response: As we mentioned above, Fig.5 was mistakenly appended to the manuscript and different from the source file uploaded separately. In our revised version, Fig.5d shows that both cholesterol and BV enhance the thermal stability of ABCB10 WT or N229A. Incubation with two compounds simultaneously displays synergic effect.

7. *Page 12 line 20, the authors speculate that their structure of ABCB10 in the absence of BV may represent the "real" apo form. One could also easily make the argument that the conformations observed in this study may not be "real" as they report on the conformation of ABCB10 solubilized in detergent rather than a true lipid bilayer. While it is appropriate to describe differences between the apo conformation observed here, and the nucleotide free structure observed previously, I would strongly discourage the use of the word "real". In reality, the true "apo" form of ABCB10 may likely be represented by an ensemble of conformational states, which may contain both the observed "apo" and "nucleotide free" conformations.*

Response: We thank the reviewer for this great suggestion. We have removed “real” in our revised version.

8. *Figure 5C caption mentions mutants...there are no mutants reported in this figure.*

Response: We thank the reviewer for this correction. We have removed “and mutant” in our revised version.

In summary, we have addressed all comments from the reviewers. We feel that our manuscript has improved a lot after revision. We hope that you will find our revised manuscript now suitable for publication.

Sincerely,